## [Reviewer comments · Life Science Alliance]

Metformin improves RAN protein pathology, alternative splicing, and behavior phenotypes in SCA8 mice

Lisa Romano, Setsuki Tsukagoshi, Emily Davey-Osuch, Ramadan Ajredini, Kamat Manasi, Tala Ortiz, Eduardo Rijios, Nathan Bourgon, S. Elaine Ames, Timothy Garrett, John Cleary, Eric Wang, and Laura Ranum

DOI: <https://doi.org/10.26508/lsa.202503555>

Corresponding author(s): Laura Ranum, University of Florida

Review Timeline:

Submission Date:	2025-10-30
Editorial Decision:	2025-11-28
Revision Received:	2026-01-21
Editorial Decision:	2026-02-04
Revision Received:	2026-02-17
Accepted:	2026-02-19

Scientific Editor: Sarita Hebbar

Transaction Report:

November 28, 2025

Re: Life Science Alliance manuscript #LSA-2025-03555-T

Dr. Laura P.W. Ranum
University of Florida
Genetics, Cell Biology and Development
6-160 Jackson Hall
Gainesville, FL 32653

Dear Dr. Ranum,

Thank you for submitting your manuscript entitled "Metformin reduces RAN proteins and rescues molecular and behavioral phenotypes in SCA8 mice" to Life Science Alliance. The manuscript was assessed by expert reviewers, whose comments are appended to this letter.

All three reviewers have commented positively on the design and execution of your study and also on its potential value to the field. That said, they have raised a few concerns that preclude publication at this stage.

The reviewers have made suggestions to (1) expand on the introduction and discussion sections and (2) provide clarity in description of results. We agree that you must include these suggestions in some form in a revised manuscript. We encourage you to add a characterisation of polyGln aggregates in the cerebellum and information on Iba1-positive microglia in the cerebellum as suggested by Reviewer 2 and 3. If this is not possible, please include a rationale for not including this data (as raised by Reviewer 3), and explicitly state its lack thereof in the manuscript

In line with their overall assessment, we invite you to submit a revised manuscript addressing the reviewers' comments. When submitting the revision, please include a letter addressing the reviewers' comments point by point. While a rebuttal must respond to all points in some form, additional experiments to resolve these points, other than indicated above, will not be required.

I would be happy to discuss the revision in more detail via email or phone/videoconferencing. Please let me know which option you prefer, if any.

Thank you for this interesting contribution to Life Science Alliance. We hope that the comments below will prove constructive as your work progresses, and we are looking forward to receiving your revised manuscript.

Sincerely,

Sarita Hebbar, PhD
Scientific Editor
Life Science Alliance
<http://www.lsjournal.org>

- A letter addressing the reviewers' comments point by point.
- An editable version of the final text (.DOC or .DOCX) is needed for copyediting (no PDFs).

B. MANUSCRIPT ORGANIZATION AND FORMATTING:

Reviewer #1 (Comments to the Authors (Required)):

This work presents a significant preclinical contribution by robustly and multifacetedly demonstrating that metformin, a safe and widely used drug, improves molecular and behavioural phenotypes in a mouse model of SCA8. The authors provide compelling evidence that the treatment not only rescues motor deficits but also specifically reduces toxic RAN protein levels and neuroinflammation, while correcting alternative splicing abnormalities, all without altering the underlying expansion RNA levels. This positions metformin as an exceptionally promising therapeutic candidate not only for SCA8 but potentially for a broad spectrum of CAG•CTG repeat expansion disorders.

The quality and robustness of the work are consistent with the high standards I have come to expect from the lead author. That said, I have a few minor questions which I believe could be clarified through discussion:

- 1) For example, given that metformin has multiple cellular targets (e.g., AMPK pathway, PKR), do you have plans for experiments to identify which of these pathways (or a combination) is primary for the reduction of RAN proteins and the improvement of splicing in your model?
- 2) The metformin dosing regimen was designed to achieve plasma levels comparable to those in humans. Could you comment on the drug's tolerability and whether any adverse effects, particularly gastrointestinal ones which are widely reported, were observed differentially between the study groups?
- 3) The study shows a partial rescue (~20%) of the mis-splicing events. Do you have a hypothesis as to why some events are more susceptible to correction by metformin than others? Could this be due to differences in the binding affinity of proteins like MBNL or other splicing factors?
- 4) The data show improvements in RAN proteins, neuroinflammation, and splicing. Do you have evidence to propose a causal relationship between these phenomena? For instance, do you believe the reduction of RAN proteins is the primary event leading to decreased inflammation and improved splicing, or could they be parallel, independent effects of the drug?

Reviewer #2 (Comments to the Authors (Required)):

In this manuscript, Romano et al. presented a comprehensive preclinical investigation into the therapeutic potential of metformin in a SCA8 BAC transgenic mouse model. The authors demonstrated that metformin treatment improved motor behavior, reduces RAN protein aggregates, alleviates neuroinflammation, and partially rescued alternative splicing defects, without altering repeat length or RNA levels.

The study is straightforward, well-designed, rigorously executed, and clearly presented. The manuscript provided strong evidence supporting metformin as a promising therapeutic candidate, and the data are generally convincing. However, the study would benefit from deeper mechanistic insight.

1. While the study convincingly showed that metformin reduced RAN protein levels and improves splicing, the direct molecular mechanism linking metformin to RAN translation suppression remains somewhat speculative. The authors referenced the PKR

pathway based on prior work (Zu et al., 2020), but no direct evidence was provided here to confirm PKR involvement in the SCA8 model. Including data on PKR phosphorylation or other downstream effectors (e.g., AMPK) in key brain regions would strengthen the mechanistic claims.

2. The observation that metformin rescued ~20% of mis-splicing events is interesting, but the functional relevance of these splicing changes was not fully explored. Were the rescued events known to be MBNL-dependent? Including MBNL localization or expression data could help link metformin's effects to known RNA gain-of-function mechanisms.

3. Figure 2 should include the polyGln staining results and corresponding quantitative analysis for the cerebellum. Figure 3 also should include the quantitative analysis for the percentage of Iba1-positive microglia in the cerebellum.

4. The legends for Figure 3 and Figure EV3 should explicitly state which brain region was shown in the representative staining images (e.g., cerebellum or brainstem).

Reviewer #3 (Comments to the Authors (Required)):

The manuscript by Romano et al describes the application of the anti-diabetic drug in mouse model for SCA8. While metformin has been used in other repeat expansion disease models for example in Huntington's disease models or models of myotonic dystrophy type 1, SCA8 models have not been investigated in response to metformin treatment yet.

The authors used a well-established animal model for their preclinical testing of metformin in SCA8. The treatment resulted in a reduction of RAN-proteins, in decreased missplicing, reduced neuroinflammation, as well as in an improvement of the motor phenotype. The study is well-designed and the experimental part includes analysis of different tests to assess the motor phenotype in combination with IHC and RNA expression analysis. The conclusions of the paper are justified based on the presented data.

I only have a few minor points that the authors may want to address:

In the abstract there is a comma missing between „Huntington's disease (HD)" and „Fuch's endothelial corneal dystrophy".

In the introduction the abbreviation ALS/FTD is not introduced.

Figure 2: Why was the cerebellum not tested for polyGln aggregates?

Figure 3: I would change the label of the upper left IHC (currently „1C2") to „polyGln 1C2". Moreover, I would include the brain region that was analyzed via IHC in the picture (maybe as header).

Discussion: what is the proposed mechanism by which metformin decreases the translation of RAN-proteins? Is this via mTOR?

We thank the reviewers for their careful and insightful comments. We address each of the reviewer concerns below.

Reviewer #1 (Comments to the Authors (Required)):

This work presents a significant preclinical contribution by robustly and multifacetedly demonstrating that metformin, a safe and widely used drug, improves molecular and behavioural phenotypes in a mouse model of SCA8. The authors provide compelling evidence that the treatment not only rescues motor deficits but also specifically reduces toxic RAN protein levels and neuroinflammation, while correcting alternative splicing abnormalities, all without altering the underlying expansion RNA levels. This positions metformin as an exceptionally promising therapeutic candidate not only for SCA8 but potentially for a broad spectrum of CAG•CTG repeat expansion disorders. The quality and robustness of the work are consistent with the high standards I have come to expect from the lead author. That said, I have a few minor questions which I believe could be clarified through discussion:

We thank reviewer#1 for the positive comments.

1) For example, given that metformin has multiple cellular targets (e.g., AMPK pathway, PKR), do you have plans for experiments to identify which of these pathways (or a combination) is primary for the reduction of RAN proteins and the improvement of splicing in your model?

We thank the reviewer for this suggestion. We now more broadly discuss the AMPK and PKR pathways that are known to be affected by metformin. We also discuss potential future experiments that will lead to a better understanding of the pathways and the role that RAN proteins in disease, including targeting the PKR pathway or RAN proteins themselves.

Revised text discussion page 11: *“Our SCA8 data raise the possibility that, decreasing RAN protein aggregate burden through the use of metformin reduces neuroinflammation in SCA8. Alternatively, metformin may reduce neuroinflammation by activating the AMP-activated protein kinase (AMPK) pathway^{13,63,64}. In C9orf72 ALS/FTD cells and mice, reducing the levels of pPKR using either a dominant negative PKR(K296R) or metformin decrease RAN protein levels¹². Future research aimed at blocking the PKR pathway or targeting SCA8 RAN proteins will lead to a better molecular understanding of how metformin improves disease in SCA8 mice.”*

Revised text discussion page 11-12: We now discuss possible thoughts to reduce RNA missplicing in DM1 and have also added a sentence about future mechanistic work needed in SCA8 to understand how metformin improves SCA8 missplicing.

“In one DM1 study, metformin was shown to have beneficial effects on selected alternative splicing abnormalities through a mechanism in which activation of AMPK downregulates the levels of the cold-shock RNA-binding motif protein 3 (RBM3)⁴⁴. ”

and

“Additional work will be needed to understand how metformin improves SCA8 missplicing abnormalities and if metformin has direct or indirect effects on one or more RNA splicing factors.

2) The metformin dosing regimen was designed to achieve plasma levels comparable to those in humans. Could you comment on the drug's tolerability and whether any adverse effects, particularly gastrointestinal ones which are widely reported, were observed differentially between the study groups?

Revised text discussion page 10-11: *We have edited the text accordingly: “An advantage of metformin treatment, as shown in this study, is that it improved both RAN and RNA splicing phenotypes without changing the levels or the balance of sense or antisense expansion RNAs. Additionally, metformin has a well-established safety record, making it a relatively safe approach for reducing RAN proteins. In patients, known adverse events of metformin range from common, mild gastrointestinal issues to rare, but serious lactic acidosis which results in severe lethargy. In our mouse study, we did not observe signs of gastrointestinal distress or lethargy in animals treated with metformin.”*

3) The study shows a partial rescue (~20%) of the mis-splicing events. Do you have a hypothesis as to why some events are more susceptible to correction by metformin than others? Could this be due to differences in the binding affinity of proteins like MBNL or other splicing factors?

We agree that metformin's partial rescue of alternative splicing in SCA8 could result from direct or indirect effects of metformin on one or more RNA splicing factors. We have now added the following sentence to address this point.

Revised text discussion page 12: *“Additional work will be needed to understand how metformin improves SCA8 missplicing abnormalities and if metformin has direct or indirect effects on one or more RNA splicing factors.”*

4) The data show improvements in RAN proteins, neuroinflammation, and splicing. Do you have evidence to propose a causal relationship between these phenomena? For instance, do you believe the reduction of RAN proteins is the primary event leading to decreased inflammation and improved splicing, or could they be parallel, independent effects of the drug?

Additional work will be needed to understand the isolated effects of RAN protein lowering versus effects on neuroinflammation. In our C9orf72 BAC transgenic mice, lowering RAN proteins with an antibody targeting RAN proteins themselves decreases neuroinflammation but teasing apart what happens in SCA8 will require additional mechanistic studies.

Reviewer #2 (Comments to the Authors (Required)):

In this manuscript, Romano et al. presented a comprehensive preclinical investigation into the therapeutic potential of metformin in a SCA8 BAC transgenic mouse model. The authors demonstrated that metformin treatment improved motor behavior, reduces RAN protein aggregates, alleviates neuroinflammation, and partially rescued alternative splicing defects, without altering repeat length or RNA levels.

The study is straightforward, well-designed, rigorously executed, and clearly presented. The manuscript provided strong evidence supporting metformin as a promising therapeutic candidate, and the data are generally convincing. However, the study would benefit from deeper mechanistic insight.

We thank Reviewer #2 for the helpful suggestions and appreciate the interest in the mechanistic aspects of metformin action. We provide our responses to the proposed analysis, changes, and corrections below.

1. While the study convincingly showed that metformin reduced RAN protein levels and improves splicing, the direct molecular mechanism linking metformin to RAN translation suppression remains somewhat speculative. The authors referenced the PKR pathway based on prior work (Zu et al., 2020), but no direct evidence was provided here to confirm PKR involvement in the SCA8 model. Including data on PKR phosphorylation or other downstream effectors (e.g., AMPK) in key brain regions would strengthen the mechanistic claims.

In the present study, we don't conclude that PKR inhibition is the mechanism responsible of clearing RAN protein aggregates. We know from previous studies that metformin is a PKR inhibitor, but we don't explore this aspect in the present study. These studies will require detailed, in-depth experiments that are outside the scope of this manuscript and will be addressed in a separate study.

2. The observation that metformin rescued ~20% of mis-splicing events is interesting, but the functional relevance of these splicing changes was not fully explored. Were the

rescued events known to be MBNL-dependent? Including MBNL localization or expression data could help link metformin's effects to known RNA gain-of-function mechanisms.

We know from other studies that, as in DM1, SCA8 CUG expansion RNAs form foci that sequester Mbnl proteins and lead to alternative splicing changes. However, the splicing abnormalities reported in this study are novel and will require additional experimental analyses beyond the scope of this study. However, we agree that focusing on Mbnl dependency and expression level will be interesting to explore in more detail in future studies.

3. Figure 2 should include the polyGln staining results and corresponding quantitative analysis for the cerebellum. Figure 3 also should include the quantitative analysis for the percentage of Iba1-positive microglia in the cerebellum.

As requested, we have performed additional experiments showing polyGln staining in cerebellar Purkinje cells and that metformin treatment reduces the number of polyGln aggregate in the cerebellar Purkinje cells ($p=0.0357$, Fig. 2J, K, L).

Revised text results page 6: *“In SCA8 mouse cerebellum, we quantified the number of aggregates in lobules VIa, VII, VIII, and IX (Fig. 2J). Similar to the brainstem, the cerebellum showed a reduced number of polyGln aggregates in metformin treated SCA8 mice ($p=0.0357$, Fig. 2K, L).”*

Revised text results page 7: *“Similarly, Iba1 staining in the cerebellum was decreased in animals treated with metformin ($p=0.0243$, Fig. 3E).”* To include the Iba1 results in the main figures, the quantification of sense and antisense transcripts and DNA repeat sizing has been moved to Fig.S2E, J, K.

4. The legends for Figure 3 and Figure EV3 should explicitly state which brain region was shown in the representative staining images (e.g., cerebellum or brainstem).

We have now added the brainstem as a region of the brain shown in the representative staining image.

Reviewer #3 (Comments to the Authors (Required)):

The manuscript by Romano et al describes the application of the anti-diabetic drug in mouse model for SCA8. While metformin has been used in other repeat expansion disease models for example in Huntington's disease models or models of myotonic dystrophy type 1, SCA8 models have not been investigated in response to metformin treatment yet. The authors used a well-established animal model for their preclinical testing of metformin

in SCA8. The treatment resulted in a reduction of RAN-proteins, in decreased missplicing, reduced neuroinflammation, as well as in an improvement of the motor phenotype. The study is well-designed and the experimental part includes analysis of different tests to assess the motor phenotype in combination with IHC and RNA expression analysis. The conclusions of the paper are justified based on the presented data.

I only have a few minor points that the authors may want to address:

We thank reviewer #3 for the thoughtful comments and insightful suggestions to strengthen our manuscript. We address each aspect below:

1. In the abstract there is a comma missing between „Huntington's disease (HD)" and „Fuch's endothelial corneal dystrophy".

This error has now been corrected.

In the introduction, the abbreviation ALS/FTD is not introduced.

We now define the ALS/FTD abbreviation in the introduction.

Figure 2: Why was the cerebellum not tested for polyGln aggregates? We now show immunohistochemical staining of the cerebellum for polyGln aggregates and have quantified our results, which show metformin treatment reduces polyGln aggregates in the cerebellum ($p=0.0357$, Fig. 2J, K, L). Please also see response to Reviewer #2, point #3.

Figure 3: I would change the label of the upper left IHC (currently „1C2") to „polyGln 1C2". Moreover, I would include the brain region that was analyzed via IHC in the picture (maybe as a header).

We have now changed the label in Figure 3 from 1C2 to polyGln 1C2, as suggested, and added the brain region shown in the representative images to the left of panel 3A. We also now similarly label Supplementary Figure 3 with “polyGln 1C2”.

Discussion: what is the proposed mechanism by which metformin decreases the translation of RAN-proteins? Is this via mTOR?

Additional research will be needed to elucidate the mechanisms by which metformin reduces RAN protein levels and whether the mTOR pathway is involved.

February 4, 2026

RE: Life Science Alliance Manuscript #LSA-2025-03555-TR

Dr. Laura P.W. Ranum
University of Florida
Genetics, Cell Biology and Development
210 Cancer Genetics Research Complex
2033 Mowry Road
Gainesville, FL 32610

Dear Dr. Ranum,

Thank you for submitting your revised manuscript entitled "Metformin reduces RAN proteins and rescues molecular and behavioral phenotypes in SCA8 mice".

Your manuscript was evaluated by all the original reviewers whose comments are appended below. As you will read, the reviewers are consistent in their views that the revised manuscript satisfactorily addresses their previous concerns.

In line with the reviewers' evaluation, we would be happy to publish your paper in Life Science Alliance pending final revisions necessary to meet our formatting guidelines.

MANUSCRIPT ORGANIZATION AND FORMATTING:

To avoid unnecessary delays in the acceptance and publication of your paper, please read the following information carefully. Full guidelines are available on our Instructions for Authors page, <https://www.life-science-alliance.org/authors>

- The titles in both the system and the manuscript file must be consistent with each other.
- Thank you for providing a 'Data Availability' Statement. Please include a repository name and persistent identifier for the RNA-seq data.
- Please provide scale bar information in the associated legend for Figures 2L, S2B, S2D.
- Please include details on Metformin (original source) in the section on Metformin administration.
- Please include objective details (name, type, N.A.) in the description of imaging (histology and fluorescent imaging).
- Please remove the section titled, "e-toc Summary" from the manuscript text file. It can still be included as the summary blurb.
- Please upload all figure files as individual ones, including the supplementary figure files; all figure legends should only appear in the main manuscript file.
- Please add your main, supplementary figure, and table legends to the main manuscript text after the references section.
- Please add the X and Bluesky handles of your host institute/organisation, as well as your own and/or one of the authors, in our system.
- Please be sure that all authors are mentioned in the Authors' Contribution section of the manuscript and that their initials match their names.
- Please rename "Competing Interest" to "Conflict of Interest."
- Please note that panels in figures must be labeled alphabetically - please correct this in figure S2, its legend, and call-outs in the manuscript text.
- Please label the references section as "References."
- Please add a callout for Figure S4F to your main manuscript text.
- Please be sure that the authorship listing and order is correct.

LSA encourages authors to provide a 30-60 second video where the study is briefly explained. We will use these videos on social media to promote the published paper and the presenting author (for examples, see <https://docs.google.com/document/d/1-UWCfbE4pGcDdcgzcmiuJI2XMBJnxKYeqRvLLrLSo8s/edit?usp=sharing>). Corresponding or first-authors are welcome to submit the video. Please submit only one video per manuscript. The video can be emailed to contact@life-science-alliance.org

FINAL FILES:

The following items are required for acceptance.

The license to publish form must be signed before your manuscript can be sent to production. A link to the license to publish form will be available to the corresponding author only. Please take a moment to check your funder requirements.

Thank you for your attention to these final processing requirements. Please revise and format the manuscript and upload materials as soon as you are able.

Thank you for this interesting contribution to the literature. We look forward to publishing your paper in Life Science Alliance.

Sincerely,

Sarita Hebbar, PhD
Scientific Editor
Life Science Alliance
<http://www.lsajournal.org>

Reviewer #1 (Comments to the Authors (Required)):

I thank the authors for their thoughtful and comprehensive responses to my queries. The manuscript has been much improved by the revisions and is now, in my opinion, ready for acceptance by Life Science Alliance.

Reviewer #2 (Comments to the Authors (Required)):

The authors have addressed all my concerns.

Reviewer #3 (Comments to the Authors (Required)):

The authors have addressed my points of criticism and responded to my satisfaction. I endorse the publication of the manuscript.

February 19, 2026

RE: Life Science Alliance Manuscript #LSA-2025-03555-TRR

Dr. Laura P.W. Ranum
University of Florida
Genetics, Cell Biology and Development
210 Cancer Genetics Research Complex
2033 Mowry Road
Gainesville, FL 32610

Dear Dr. Ranum,

Thank you for submitting your Research Article entitled "Metformin improves RAN protein pathology, alternative splicing, and behavior phenotypes in SCA8 mice". It is a pleasure to let you know that your manuscript is now accepted for publication in Life Science Alliance. Congratulations on this interesting work.

DISTRIBUTION OF MATERIALS:

Again, congratulations on a very nice paper. I hope you found the review process to be constructive and are pleased with how the manuscript was handled editorially. We look forward to future exciting submissions from your lab.

Sincerely,

Sarita Hebbar, PhD
Scientific Editor
Life Science Alliance
<http://www.lsajournal.org>